# Early circulating tumor DNA dynamics and clonal selection with palbociclib and fulvestrant for breast cancer

Ben O'Leary [1,2], Sarah Hrebien[1], James P. Morden[3], Matthew Beaney[1], Charlotte Fribbens[1,2], Xin Huang[4], Yuan Liu[4], Cynthia Huang Bartlett[4], Maria Koehler[4], Massimo Cristofanilli[5], Isaac Garcia-Murillas[1], Judith M. Bliss[3] & Nicholas C. Turner[1,2]

CDK4/6 inhibition substantially improves progression-free survival (PFS) for women with advanced estrogen receptor-positive breast cancer, although there are no predictive bio-markers. Early changes in circulating tumor DNA (ctDNA) level may provide early response prediction, but the impact of tumor heterogeneity is unknown. Here we use plasma samples from patients in the randomized phase III PALOMA-3 study of CDK4/6 inhibitor palbociclib and fulvestrant for women with advanced breast cancer and show that relative change in *PIK3CA* ctDNA level after 15 days treatment strongly predicts PFS on palbociclib and ful-vestrant (hazard ratio 3.94, log-rank $p = 0.0013$). *ESR1* mutations selected by prior hormone therapy are shown to be frequently sub clonal, with *ESR1* ctDNA dynamics offering limited prediction of clinical outcome. These results suggest that early ctDNA dynamics may provide a robust biomarker for CDK4/6 inhibitors, with early ctDNA dynamics demonstrating divergent response of tumor sub clones to treatment.

[1] Breast Cancer Now Research Centre, The Institute of Cancer Research, Fulham Rd, London SW3 6JB, UK. [2] Breast Unit, Royal Marsden Hospital, London SW3 6JJ, UK. [3] The Institute of Cancer Research Clinical Trials and Statistics Unit, London SM2 5NG, UK. [4] Pfizer, 235 E 42nd St, New York, NY 10017, USA. [5] Robert H Lurie Comprehensive Cancer Center, Feinberg School of Medicine, 675 N St. Clair, Chicago, IL, 60611, USA. James P. Morden is deceased. Correspondence and requests for materials should be addressed to N.C.T. (email: nicholas.turner@icr.ac.uk)

Circulating tumor DNA (ctDNA) can be detected in the cell-free DNA (cfDNA) of patients with cancer by identifying genomic aberrations. The levels of ctDNA fluctuate with treatment during cancer therapy, broadly correlating with disease response and relapse across a wide range of cancers and treatments and offering the potential for non-invasive monitoring of disease status in both the metastatic and early settings[1–5]. Circulating tumor DNA is a surrogate for tumor biopsy in assessment of driver mutations such as *EGFR* in lung cancer[6] and allows assessment of clonal evolution over time in patients with multiple metastases[7]. There are now early data that suggest that ctDNA dynamics in the early stages of treatment could be used to predict treatment outcome before tumor response can be assessed through the conventional means of size on imaging or a change in clinical symptoms or signs. Early ctDNA dynamics predict outcome on chemotherapy for colon cancer[8, 9].

These studies have shown that early ctDNA dynamics could potentially allow a real-time, individualized assessment of a treatment's efficacy, allowing treatment adaptation with either an early switch away from an ineffective therapy or addition of further therapy in a group identified early as being resistant to therapy. However there are no data from large, homogenously treated phase III trials[10–13], that allow a robust determination of clinical validity for ctDNA dynamics in assessing response to targeted therapies.

The potential for ctDNA dynamics to predict long-term outcome is based on two assumptions. The first assumption is that a fall in measured ctDNA corresponds to a functional effect of treatment on the cancer, and that short-term effects of treatment will predict long-term clinical outcome. The exact mechanisms by which ctDNA enters the circulation are incompletely understood[14], but are likely in part a function of tumor cell death and turn-over given the observed association between ctDNA and tumor volume[15, 16]. Specific treatment effects on these processes may be important in relating early ctDNA dynamics to clinical outcome.

The second assumption is that an early measurement of ctDNA can provide an adequate surrogate of the overall, aggregate response of a patient's cancer to treatment. Intra-tumoral genetic heterogeneity presents a challenge to targeted therapy response, as sub clones may have divergent response to therapy. An example of this phenomenon is the occurrence of mutations in the estrogen receptor gene, *ESR1*, in breast cancer that result in constitutive activation of the estrogen receptor[17, 18]. *ESR1* mutations evolve in response to prior aromatase inhibitor therapy for advanced breast cancer in as many as a third of patients[17, 19–22]. Prior data has suggested that *ESR1* mutations may frequently be sub clonal in the resistant cancer[23], yet the extent to which this sub clonality effects response to therapy is unknown. The selective estrogen degrader fulvestrant has activity in vitro against cancers with *ESR1* mutations, although higher concentrations are required to inhibit mutant ER than wild-type ER and it is unknown if the clinical schedules of fulvestrant achieve the doses required to degrade mutant ER[17].

We undertook an analysis of the PALOMA-3 trial to assess the utility of early ctDNA dynamics in predicting outcome on CDK4/6 inhibitors, and to investigate the implications of tumor heterogeneity. CDK4/6 inhibitors have substantial efficacy in the treatment of advanced estrogen (ER) receptor-positive HER2-negative breast cancer, the most common subtype of breast cancer. Palbociclib, a CDK4/6 inhibitor, increases progression-free survival when added to the endocrine therapies letrozole[24] and fulvestrant[25], and ribociclib increases progression-free survival when added to letrozole[26]. The PALOMA-3 trial compared palbociclib plus fulvestrant with fulvestrant plus placebo in a 2:1 ratio in advanced breast cancer that had previously progressed on endocrine treatment, demonstrating an improvement in progression-free survival from 4.6 months to 9.5 months with the addition of palbociclib to fulvestrant[25]. However, studies have been unable to identify which ER-positive breast cancers benefit from CDK4/6 inhibition[27], and there is a pressing need to identify biomarker approaches to identify which patients derive long-term benefit.

Here we show that early ctDNA dynamics in the commonly truncal mutations in *PIK3CA* predict sensitivity to palbociclib. In contrast we show that *ESR1* mutations are commonly sub clonal and weak predictors of outcome, with early ctDNA dynamics anticipating clonal selection of *ESR1* mutations by therapy.

## Results

**Early plasma dynamics of *PIK3CA* mutations.** Sequential plasma samples were collected at baseline, cycle 1 day 15, and at progression in the PALOMA-3 study to assess whether early dynamic changes in ctDNA could predict progression-free survival (PFS) on palbociclib. Of the 521 patients recruited to the study, 459 baseline samples were available for DNA extraction of which 455 were analyzed with multiplex digital PCR assays for hotspot mutations in exons 9 and 20 of *PIK3CA* (E542K, E545K, H1047R and H1047L) (Fig. 1a). This analysis identified 100 cases (22.0%) with a *PIK3CA* mutation (Supplementary Fig. 1), a similar proportion to that seen in The Cancer Genome Atlas for these mutations in primary disease (22.1%)[28, 29], *PIK3CA* mutations being predominantly truncal events in breast cancer[30–34]. Four cases in the PALOMA-3 set demonstrated two *PIK3CA* mutations (4%), all of which featured an E545K clone, demonstrating a sub clonal element of *PIK3CA* mutation (Supplementary Fig. 2). Similarly, in TCGA data only 0.7% (1/152) of ER+/HER2- *PIK3CA* mutant patients had double mutations detected among these *PIK3CA* mutations[35].

Matched day 15 samples were available for 73 patients (Fig. 1a). There was a statistically significant decline in copies/ml for both mutant (median relative change 0.076, $p < 0.0001$, Wilcoxon signed-rank test) and wild-type alleles (median relative change 0.542, $p < 0.0001$, Wilcoxon signed-rank test) of *PIK3CA* (Fig. 1b). Wild-type allele change reflected changes in total cell-free DNA, with a substantially more marked fall for mutant *PIK3CA* as a marker of ctDNA ($p < 0.0001$, Wilcoxon signed-rank test; Fig. 1b, see also Supplementary Fig. 3).

We defined the "circulating DNA ratio" (CDR) as the ratio of mutation abundance (mutant copies/ml) on treatment relative to baseline, $CDR_{15}$ denoting the ratio of cycle 1 day 15 to baseline. Patients randomized to palbociclib plus fulvestrant had a lower *PIK3CA* $CDR_{15}$ compared to fulvestrant plus placebo ($p < 0.0001$, Mann–Whitney test; Fig. 1c) indicating larger reduction in ctDNA abundance. All patients on palbociclib (52/73) had a $CDR_{15} < 1$, indicating a fall in circulating tumor DNA. These data demonstrate that the anti-proliferative effects of palbociclib result in a rapid fall in ctDNA levels by day 15, and that *PIK3CA* $CDR_{15}$ assessment anticipated the improved PFS seen with palbociclib in the PALOMA-3 trial[25]. Reduction of wild-type copies was predominantly in the palbociclib treatment group (median $CDR_{15}$ 0.36 v 0.85, palbociclib vs placebo, $p = 0.0005$, Mann–Whitney test, Supplementary Fig. 4), likely reflecting both reduction in wild-type alleles released from tumor cells and the cytostatic effect of palbociclib on hemopoietic cells.

**Early plasma dynamics of *ESR1* mutations.** We then assessed whether early ctDNA dynamics were different with evolved *ESR1* mutations that are frequently sub clonal[20]. Of the baseline PALOMA-3 samples, 445 (85.4% of enrolled patients) were analyzed for common mutations in the ligand-binging domain of

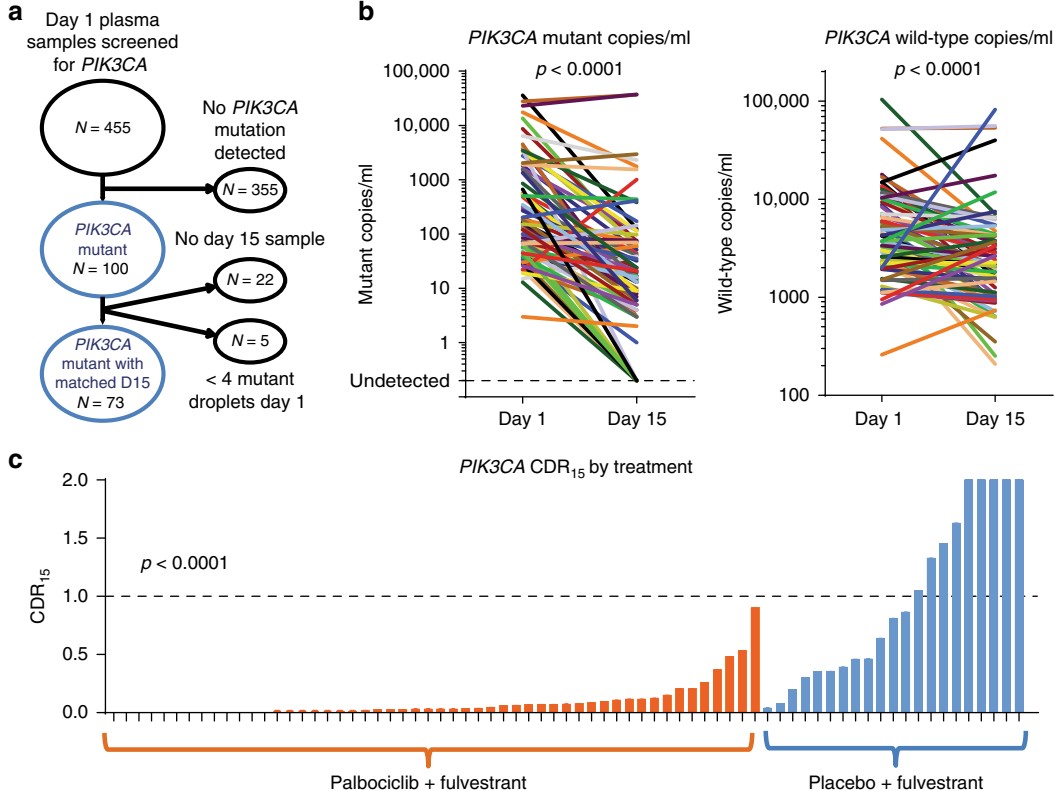

**Fig. 1** Palbociclib and fulvestrant suppresses ctDNA after two weeks of treatment. **a** CONSORT diagram of plasma samples analyzed from PALOMA-3 for *PIK3CA* mutation with droplet digital PCR assays. **b** Dynamics of *PIK3CA* mutant and wild-type DNA copies/ml between day 1 and day 15 of treatment, $n = 73$, median day15:day1 ratio for mutant 0.076, median ratio for wild-type 0.54, $p < 0.0001$, Wilcoxon signed-rank test. **c** Circulating DNA ratio day15 copies/ml relative to day 1 copies/ml ($CDR_{15}$) for circulating *PIK3CA* mutation split by treatment. *p* value Mann–Whitney test comparison between treatments. Lines are at median

*ESR1* (E380Q, S463P, L536R, Y537S, Y537C, Y537N, and D538G) using multiplex droplet digital PCR assays. Baseline *ESR1* mutations were identified in 114 (25.6%) patients (Fig. 2a), 33 (28.9%) of these were polyclonal, the majority of polyclonal samples featuring a D538G mutation (29/33) (Supplementary Fig. 5 and Supplementary Fig. 6). Of the 114 *ESR1* mutant baseline samples, matched day 15 was available in 65 patients (Fig. 2a). As with *PIK3CA*, both the aggregate *ESR1* mutant copies (median relative change 0.022, $p < 0.0001$, Wilcoxon signed-rank test) and wild-type allele (median relative change 0.21, $p < 0.0001$, Wilcoxon signed-rank test) were significantly lower at day 15 (Fig. 2b; Supplementary Fig. 7) and significantly lower with palbociclib (Fig. 2c; $p = 0.034$, Mann–Whitney test). However, in the fulvestrant and placebo group, patients with *ESR1* mutations had substantially greater suppression of mutant *ESR1* ctDNA than patients with *PIK3CA* mutations (median $CDR_{15}$ *ESR1* 0.044 vs *PIK3CA* 0.82, $p < 0.0001$, Mann–Whitney test; Fig. 2d), suggestive of a differential response. There was also a non-significant trend towards greater suppression on palbociclib and fulvestrant (median $CDR_{15}$ *ESR1* 0.014 vs *PIK3CA* 0.034, $p = 0.0532$, Mann–Whitney test; Fig. 2e).

**Early *PIK3CA* dynamics predict outcome on palbociclib.** We next investigated whether $CDR_{15}$ assessment could predict long-term outcome for patients treated with palbociclib, relating *PIK3CA* and *ESR1* $CDR_{15}$ to PFS using a Cox proportional hazards model. Patients with *PIK3CA* $CDR_{15}$ above the median value of 0.034 had inferior PFS compared to those below median (hazard ratio 3.94, 95% CI 1.61–9.64, log-rank $p = 0.0013$;

Fig. 3a). In contrast, *ESR1* $CDR_{15}$ using the median failed to identify a significant relationship with PFS (Fig. 3b). A single cut-point for predicting PFS on palbociclib and fulvestrant using the *PIK3CA* $CDR_{15}$ was optimized using Harrell's c-index, with *p* values corrected using the Benjamini–Hochberg method to adjust for multiple comparisons (Supplementary Fig. 8). Using this cut-point (Fig. 3b), patients with a high $CDR_{15}$ had a median PFS of 4.1 months (95% CI 3.6–5.5) and patients with a low, suppressed, $CDR_{15}$ had a median PFS of 11.2 months (95% CI 11.1–undefined), hazard ratio 4.92 (95% CI 1.98–12.26, log-rank test $p = 0.0002$, $q = 0.007$). It was not possible to optimize a statistically significant single cut-point in *ESR1* $CDR_{15}$ with Harrell's c-index (Supplementary Fig. 9) with a Benjamini–Hochberg correction (Fig. 3d; $q = 0.15$). Therefore, *PIK3CA* ctDNA dynamics after two weeks of palbociclib and fulvestrant predicted long-term clinical outcome on palbociclib, while *ESR1* ctDNA dynamics were less predictive.

Baseline ctDNA burden has been linked to prognosis in some prior studies[9]. The baseline level of *PIK3CA* ctDNA did not predict PFS on palbociclib and fulvestrant (above vs below median HR = 1.22, 95% CI 0.606–2.43, log-rank $p = 0.582$), and there was no correlation between baseline mutant copies/ml and $CDR_{15}$, (Supplementary Fig. 10. *PIK3CA* $CDR_{15}$ remained significant in a multivariable analysis of prognostic clinical and pathological features on palbociclib (Supplementary Fig. 11). Analysis of *PIK3CA* and *ESR1* $CDR_{15}$ in patients on fulvestrant and placebo was performed but was limited by low patient numbers (HR = 2.54, 95% CI 0.89–7.25, log-rank $p = 0.07$ Supplementary Figs. 12 and 13).

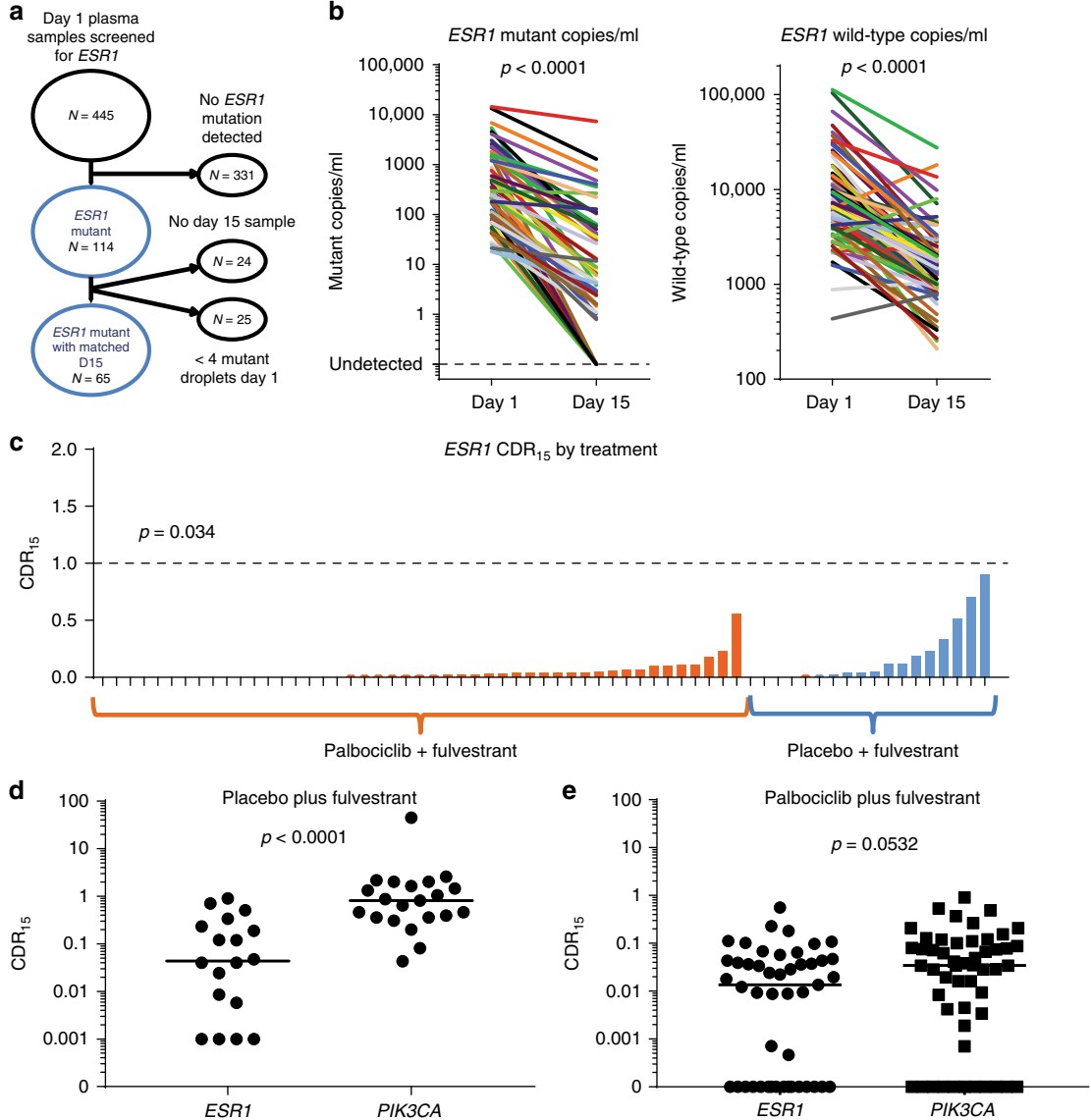

**Fig. 2** *ESR1* mutant sub clones have different early ctDNA dynamics to *PIK3CA* mutations. **a** CONSORT diagram for samples from PALOMA-3 used for the *ESR1* mutation analysis. **b** Dynamics of *ESR1* mutant and wild-type DNA copies/ml between day 1 and day 15 of treatment, $n = 65$, median ratio for mutant 0.022, median ratio for wild-type 0.21, $p$ value Wilcoxon signed-rank test. **c** Mutant *ESR1* $CDR_{15}$ split by treatment. $p$ value Mann–Whitney test comparing treatments. **d** $CDR_{15}$ for *PIK3CA* mutations vs *ESR1* mutations in patients randomized to fulvestrant plus placebo. $p$ value Mann–Whitney test. Line at median. **e** $CDR_{15}$ for *PIK3CA* mutations vs *ESR1* mutations in patients receiving palbociclib plus fulvestrant. $p$ value Mann–Whitney test. Line at median

**ESR1 mutant sub clones do not reflect outcome on fulvestrant.** The early ctDNA dynamics data suggested that *ESR1* mutant clones demonstrated greater suppression of $CDR_{15}$ on fulvestrant, potentially providing evidence that fulvestrant has potency against *ESR1* mutant tumors in the clinic[36]. However, this is at odds with prior data that suggests that baseline *ESR1* mutations did not predict for improved PFS on fulvestrant[21]. We analyzed the effect of baseline *ESR1* mutation detection on outcome on fulvestrant and placebo in an updated analysis of 151 patients[21]. Patients with baseline *ESR1* mutation detected had a statistically significantly worse PFS compared to patients with baseline wild-type *ESR1* (HR 1.58, 95% CI 1.02–2.43, log-rank $p = 0.04$; Fig. 4a). We subsequently investigated possible explanations for this disconnect between the acute effects of fulvestrant on *ESR1* $CDR_{15}$ and the lack of long-term benefit assessed by PFS by examining the clonal status of *ESR1* mutations in this cohort.

Of the patients with *ESR1* mutation at baseline, 35 also had a *PIK3CA* mutation enabling clonal comparison within each patient (Fig. 4b). Confidently determining whether a mutant population is truncal is challenging in plasma due to great variability in tumor purity and variation in the copy number status of alleles. Nevertheless, the summed allele fraction of *ESR1* mutations was lower than *PIK3CA* allele fraction in 77.1% patients with both mutations (Fig. 4c), suggesting the *ESR1* mutations were predominantly sub clonal, although this analysis is limited by an incomplete analysis of the *ESR1* gene. In the 25 patients with assessable $CDR_{15}$ and dual *PIK3CA* and *ESR1* mutations, the association in $CDR_{15}$ was linear between two mutations in the majority of samples, whereas solely the *ESR1* mutant clone became undetectable in 32% (8/25) suggestive of contrasting clonal dynamics of a sub clonal *ESR1* mutation (Fig. 4d). Although the *ESR1* mutant clone becoming undetectable could in part be due to difference in level of detection between *PIK3CA* and *ESR1* mutations (Supplementary Figs. 14 and 15), *ESR1* became undetectable at a higher rate than would be expected by random sampling taking into account difference in

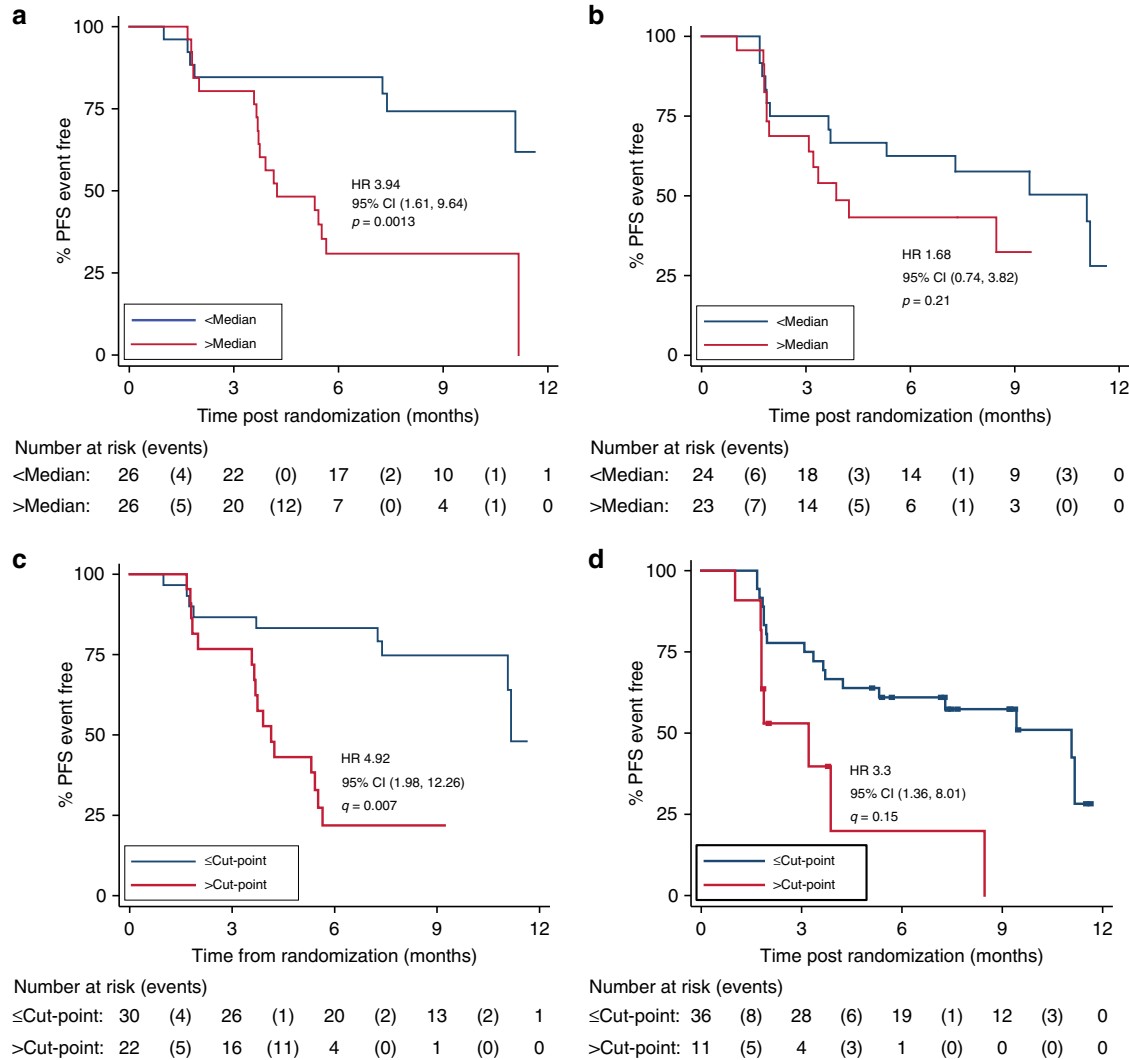

**Fig. 3** Early *PIK3CA* ctDNA dynamics predict progression-free survival (PFS) on palbociclib and fulvestrant more strongly than *ESR1* dynamics. **a** Kaplan–Meier plot for PFS of patients randomized to palbociclib and fulvestrant split by median *PIK3CA* $CDR_{15}$. **b** Kaplan–Meier plot for PFS of patients randomized to palbociclib and fulvestrant split by median *ESR1* $CDR_{15}$. **c** Kaplan–Meier plot for PFS of patients receiving palbociclib and fulvestrant split by high or low *PIK3CA* $CDR_{15}$ using an optimized cut-off calculated with Harrell's c-index. q value log-rank test corrected for false discovery with Benjamini–Hochberg. **d** Kaplan–Meier plot for PFS of patients receiving palbociclib and fulvestrant split by high or low *ESR1* $CDR_{15}$ using an optimized cut-off calculated with Harrell's c-index. q value log-rank test corrected for false discovery with Benjamini–Hochberg

level of detection (Supplementary Table 1). Baseline mutant copies/ml was not associated with being undetectable at day 15 for *ESR1* (Supplementary Fig. 16).

Taken together, these data are consistent with *ESR1* mutation representing a sub clone in a substantial fraction of patients with advanced breast cancer, poorly representing the cancer overall. This sub clone may show greater initial sensitivity to fulvestrant combinations, evident in the early ctDNA dynamics, while other sub clones may respond differently such that response of the *ESR1* mutant clone may not eventually determine clinical outcome.

**Early sub clonal dynamics predict clonal detection on relapse.** Our data suggested that evolved *ESR1* mutations are frequently sub clonal, and that these sub clones may show initial increased sensitivity to fulvestrant, and to palbociclib, as assessed by $CDR_{15}$. We next assessed whether early clonal dynamics in ctDNA could predict the clonal composition at relapse. Of the 113 samples assessed for $CDR_{15}$, with either a *PIK3CA* or *ESR1* mutation or both, 59 (52.2%) had an end of treatment plasma sample (Fig. 5a).

All except one of the cases positive for a *PIK3CA* mutation at baseline had the same mutation detectable at the end of treatment (1/37 undetectable, 2.7%), consistent with the probable truncal nature of *PIK3CA* mutations in breast cancer (Fig. 5b). In contrast, 8 of 31 patients (25.8%) with *ESR1* mutation at baseline had undetectable *ESR1* mutation at the end of treatment, significantly more than *PIK3CA* mutations (Fig. 5b; $p = 0.005$, two sample test of proportions). We then investigated whether early clonal dynamics anticipated loss of the *ESR1* clone at subsequent relapse. Five of the nine (55.6%) *ESR1* mutant subjects with undetectable ctDNA at day 15 had no detectable *ESR1* at end of treatment (Fig. 5c, d; $p = 0.027$, Fisher's exact test), suggesting early clonal dynamics predicted for clonal composition at progression. Four of the right cases with undetectable *ESR1* mutation at progression were also *PIK3CA* mutated at baseline and all of these had *PIK3CA* mutation detected at the end of treatment, indicating the observed loss of *ESR1* mutation was not an artifact of low tumor content. There was no evidence from the six cases with polyclonal *ESR1* mutations at baseline of favorable selection of one *ESR1* mutation over another (Supplementary Fig. 17).

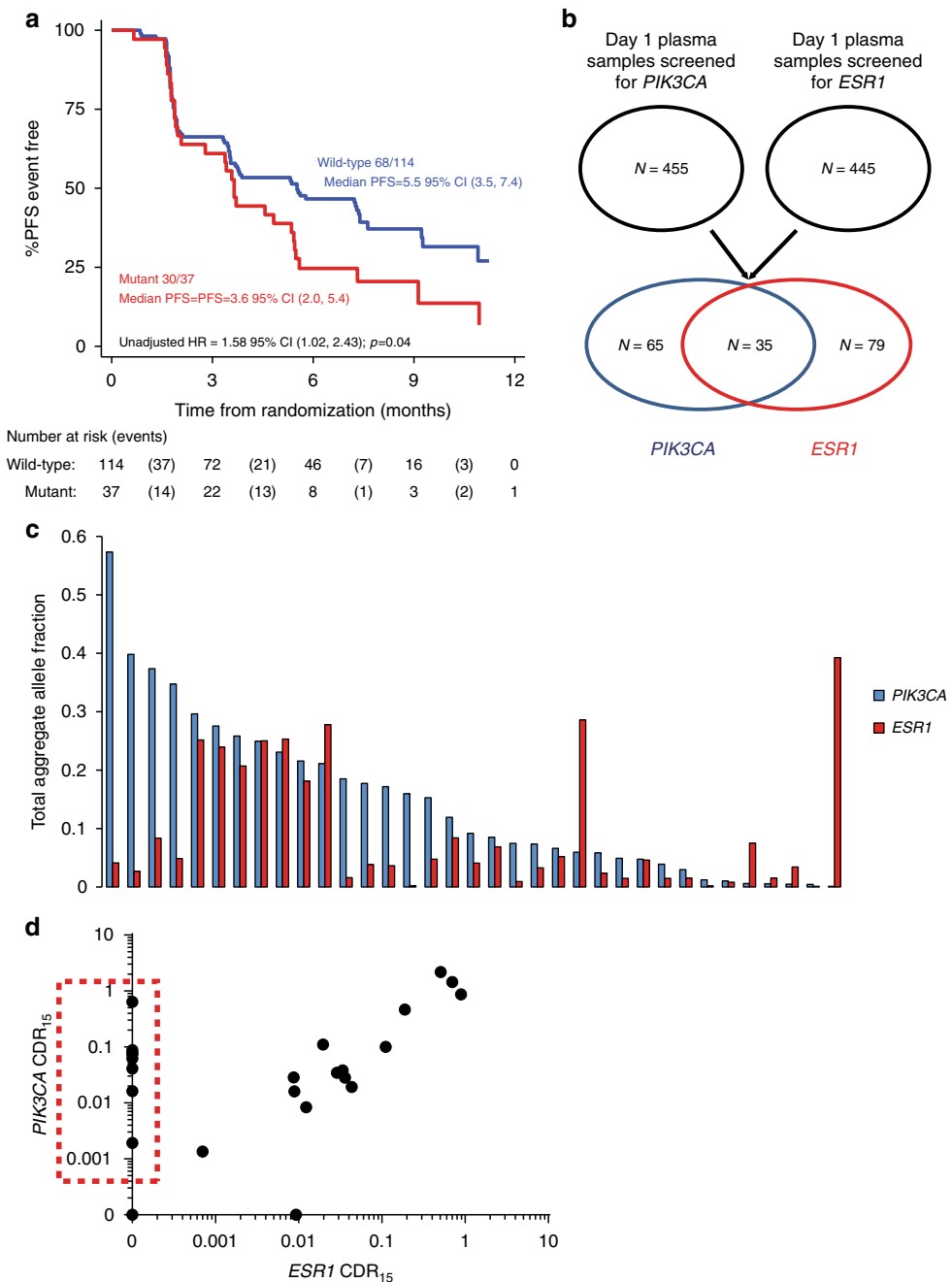

**Fig. 4** *ESR1* mutations may be sub clonal with distinct response to therapy. **a** Kaplan–Meier curves for PFS of patients with and without *ESR1* mutations in day 1 baseline plasma randomized to placebo and fulvestrant in the PALOMA-3 trial, *n* = 151, updated from ref. [17]. **b** Overlap of baseline plasma *PIK3CA* and *ESR1* mutations. **c** Baseline day 1 allele fraction comparison of aggregate *PIK3CA* mutations and aggregate *ESR1* mutations in patients with mutations detected in both genes, *n* = 35. **d** Comparison of CDR$_{15}$ for *PIK3CA* mutation and *ESR1* mutation in the same patient plasma samples, *n* = 25. The red dashed box highlights patients with loss of *ESR1* mutation in plasma after 15 days treatment where *PIK3CA* mutation remained detectable

Clearance of *ESR1* mutation at end of treatment was more frequent in patients on palbociclib and fulvestrant than those on fulvestrant and placebo (35.6% 7/20 v 9.1% 1/11, respectively) though this was not a statistically significant result (*p* = 0.2, Fisher's exact test).

## Discussion

In this analysis evaluating the predictive role of longitudinal ctDNA assessment in the PALOMA-3 study, we show that early circulating tumor DNA dynamics with truncal *PIK3CA* mutations predict sensitivity to palbociclib in advanced hormone receptor-positive breast cancer. The early dynamics of *ESR1* mutations, commonly sub clonal, did not predict sensitivity, or at best did not predict as strongly as the dynamics of *PIK3CA* mutations. Evolved *ESR1* mutant clones show an initial marked decrease in abundance to both the combination of palbociclib and fulvestrant and fulvestrant alone as assessed by early circulating tumor DNA dynamics, but this does not associate with long-term improvement in PFS on fulvestrant alone relative to patients with wild-type *ESR1*. *ESR1* mutations were more frequently undetectable at the end of treatment than *PIK3CA* mutations, and early clonal selection of *ESR1* mutations can be detected after

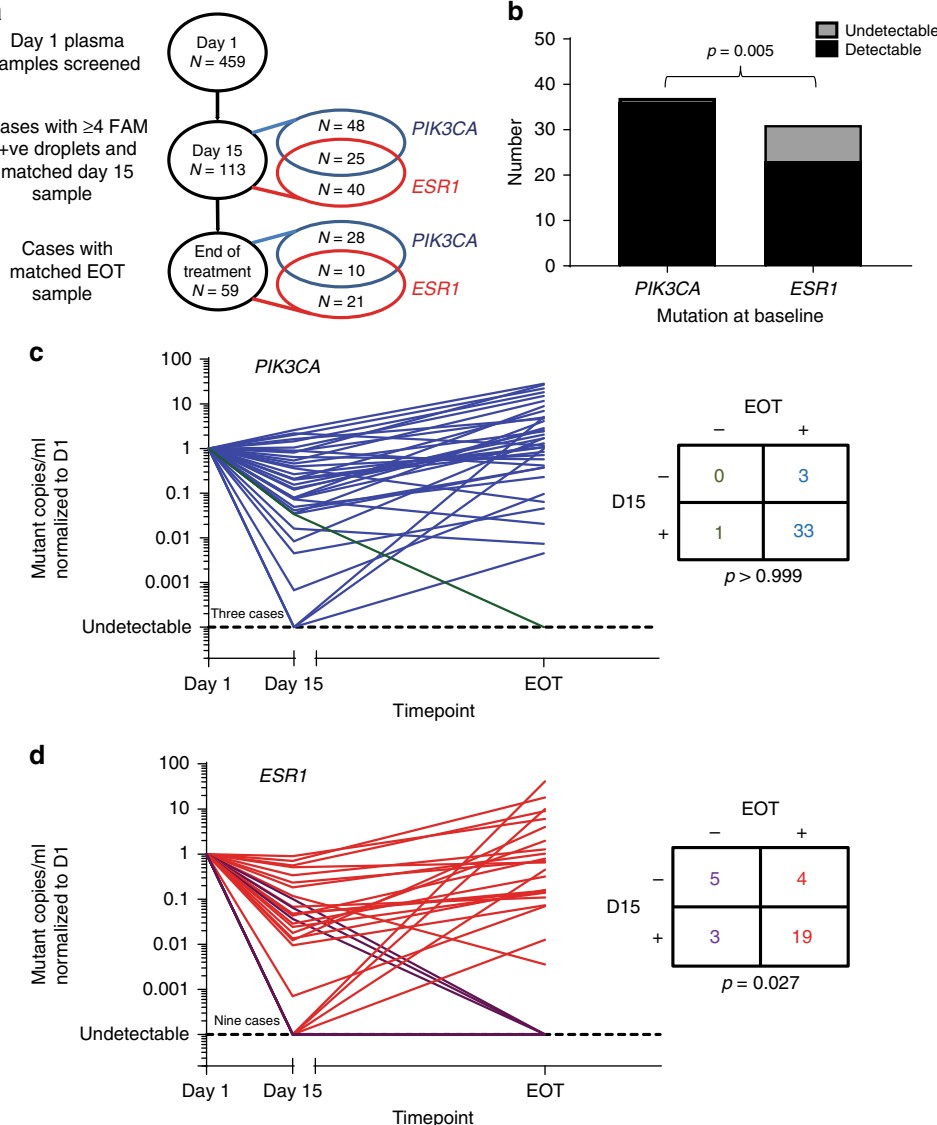

**Fig. 5** Clonal composition at relapse is anticipated by early ctDNA dynamics. **a** Samples with *PIK3CA* and *ESR1* mutations used in longitudinal clonal analysis. **b** Comparison of *PIK3CA* mutation and *ESR1* mutation detection at end of treatment. *p* value two sample *t* test. **c** Spider plot for 37 patients with *PIK3CA* mutation in longitudinal clonality analysis. Mutant copies/ml normalized to day 1. Lines are colored as for the accompanying contingency table, patients with detectable end of treatment mutation colored blue and undetectable colored green. *p* value Fisher's exact test. **d** Spider plot for 31 patients with *ESR1* mutation in longitudinal clonality analysis. Mutant copies/ml are normalized to the value at day 1. Lines are colored as for the accompanying contingency table, patients with detectable end of treatment mutation colored red and undetectable colored purple. *p* value Fisher's exact test

15 days treatment, anticipating clonal architecture at progression. In combination these observations present a potential strategy to address the impact of intra-tumor genetic heterogeneity through early assessment of ctDNA from truncal and sub clonal mutations, giving an earlier indication of outcome and rapidly identifying sub clones that will not dictate long-term outcome. These data also highlight the importance of understanding the clonal status and functional significance of genetic alterations in ctDNA when extrapolating between early ctDNA dynamics and clinical end points.

Prior research has been unable to identify tumor-based biomarkers of sensitivity to palbociclib in ER-positive breast cancer. Neither *PIK3CA* nor *ESR1* mutant status at baseline were found to predict outcome on palbociclib and fulvestrant in previous analyses of the PALOMA-3 trial[21, 25]. Amplification of *CCND1*, the gene-encoding cyclin D1, and loss of p16 were assessed prospectively in the PALOMA-1/TRIO-18 trial of palbociclib and

letrozole in first-line advanced, hormone receptor-positive breast cancer, but neither criterion identified a subgroup of better responders[37]. Functional retinoblastoma protein (Rb, encoded by the *RB1* gene) is a requirement for response to CDK4/6 inhibition in vitro, but this has not been demonstrated in the clinic, possibly in part as *RB1* aberrations are rare in ER-positive breast cancer[38]. Early on-treatment assessment of ctDNA dynamics represents the first biomarker for predicting response to palbociclib (Fig. 3). Our results here demonstrate that early suppression of ctDNA predicts long-term outcome substantially earlier than changes in tumor size, when truncal mutations such as *PIK3CA* are tracked in plasma, in the setting of palbociclib with fulvestrant in advanced hormone receptor-positive breast cancer.

Our data demonstrates that early ctDNA changes may be predictive for cytostatic therapies such as palbociclib. Cancer cells predominantly undergo apoptosis as a result of a failure to transition the S and M phases of the cell cycle. Our data suggest

cancers with incomplete cell cycle arrest on palbociclib continue to proliferate on treatment, undergoing apoptosis and/or necrosis with ongoing release of tumor DNA in the circulation. In contrast, cancers with more complete G0/G1 cell cycle arrest to palbociclib no longer undergo cell death and have more fully suppressed ctDNA. Ensuring complete cell cycle arrest by CDK4/6 inhibitor combinations is therefore shown to be key to maintaining long-term disease control by therapy.

As *PIK3CA* mutation detection at baseline is not a predictive biomarker for palbociclib and fulvestrant, in this analysis it likely represents a functionally agnostic means of assessing the presence of DNA released by the patients' cancer cells, its truncal status meaning it is found in all cancer cells within the patient. Our data illustrate the importance of assessing clonality in using early ctDNA dynamics to predict response to targeted therapies in cancer. *ESR1* mutant clones evolve in ER-positive advanced breast cancer during aromatase inhibitor treatment, with *ESR1* mutations resulting in constitutively active ER[17, 19–21]. We show that *ESR1* mutations are frequently sub clonal, (Fig. 4c, d) with high levels of polyclonality (Supplementary Fig. 6). *ESR1* mutation sub clonality means they are poorly representative of a patient's cancer overall, limiting the potential for their ctDNA dynamics to predict clinical outcome (Fig. 3b, d; Supplementary Fig. 9). Furthermore, though *ESR1* mutant clones may be acutely sensitive to fulvestrant or the combination of palbociclib and fulvestrant (Fig. 2d), this does not translate into improved PFS for patients with *ESR1* mutant cancers (Fig. 4a). In contrast to this we show that patients with *ESR1* mutations in their cancers have worse outcome on fulvestrant alone. This discrepancy could potentially be explained by cancers with *ESR1* mutations detected in plasma being genetically diverse and more likely to feature other resistant sub clones not assessed here, highlighting the potential importance of understanding heterogeneity with non-invasive assays. Previous studies of early ctDNA dynamics within the first 2 weeks of treatment have not assessed sub clonality or explored functionally distinct clones, using either a single gene or heterogeneous cohorts receiving a variety of treatments and measuring a number of different genetic aberrations[8, 9]. That a colorectal study in a heterogeneous cohort was able to show some clinically predictive value for early ctDNA dynamics may be because the majority of the mutations that were assessed were probably truncal, having been selected on the basis of prevalence in primary disease[9]. Relative early clonal dynamics could offer an approach to tackle intra-tumoral genetic heterogeneity, through identifying differences in sensitivity to treatment early, and anticipating changes in clonal composition at progression.

The early ctDNA dynamics of a truncal mutation approach could be applied to other cancer treatments with no biomarker, but a number of challenges remain. Palbociclib's cytostatic effect may be significant in generalizing this analysis to other treatments which are not cytostatic. The optimal time point for comparative assessment is also unknown. Further problems in generalizing this approach include confident selection of a truncal mutation and understanding the functional significance of sub clones, as seen here with the contrasts between *PIK3CA* and *ESR1*. Technically the best method for measuring the tumor content of plasma for early changes remains uncertain. The advantages of digital PCR, speed, sensitivity and precision at very low concentrations with absolute quantification, could be challenged by multi-region sequencing with error-correcting molecular barcodes[15, 39]. This would also allow a broader range of truncal and sub clonal aberrations to be assessed, providing an opportunity for a richer sub clonal analysis.

These results are exploratory and care must be taken with their interpretation. Strengths of this analysis include samples and outcome data being collected within a large phase III randomized

controlled trial of a relatively homogeneous cohort of patients, with samples collected specifically to analyze the potential of $CDR_{15}$ to predict PFS. Heterogeneity of unselected cohorts on a variety of treatments presents significant challenges for meaningful studies of comparative clonal dynamics. A limitation of our study is a degree of uncertainty concerning the true truncal or sub clonal status of the *PIK3CA* and *ESR1* mutations, which we infer rather than directly assess with multiple region, multiple time point biopsies. Data from archival primary tissue would be helpful in supporting the hypothesis that the majority of the *PIK3CA* mutations we observe are truncal. Analysis of TCGA data confirmed that polyclonal *PIK3CA* mutations are rare in primary disease, but the landscape of endocrine-resistant disease is less well described. A small subset of the *PIK3CA* mutations observed in the PALOMA-3 analysis may be sub clonal, as is potentially the case for the patients observed to have an *ESR1* mutation at higher allele fraction than their *PIK3CA* mutation. Another limitation is the relatively modest amount of plasma we were able to assess, although this did not substantially affect our analysis (Supplementary Fig. 14). Importantly, we also lack an independent clinical dataset to validate the *PIK3CA* cut-off for $CDR_{15}$; this would be required before this criterion could be tested for use with clinical decision-making.

It is uncertain if these results would apply to other palbociclib—hormone therapy combinations, and to other CDK4/6 inhibitors. Relatively few patients had end of treatment samples, and the frequency of loss of *ESR1* mutations at end of treatment is inaccurately estimated. Our analysis is specific to digital PCR analysis, and separate cut-offs would need to be derived for alternative ctDNA analysis techniques such as error-corrected molecular barcode sequencing[15, 39]. A further consideration is whether our optimized $CDR_{15}$ cut-off would be too stringent for clinical purposes, and whether a higher cut-off should be employed, defined through deciding on a minimum clinically acceptable median PFS benefit.

Following validation in an independent clinical cohort, these data would support trials testing the hypothesis that a change in treatment strategy based on early ctDNA dynamics may improve outcome, by switching to another modality or adding additional treatment for patients with inadequate ctDNA suppression. For example, patients with advanced hormone receptor-positive *PIK3CA* mutant breast cancer with inadequate ctDNA suppression on palbociclib and fulvestrant could benefit from early introduction of a PI3-kinase inhibitor. Further clinical trials using a validated cut-off for circulating DNA response will be required to test this approach prospectively. It may be further possible to translate this approach to allow early assessment of other treatments, and individualize assessment of response to therapy. Conceivably ctDNA response could also be used to screen agents for signs of efficacy in early phase trials. Future trials examining the early dynamics of circulating tumor DNA need to consider the clonal status of the measured aberration, and whether it is appropriate to group together aberrations that may be sub clonal and functionally distinct.

## Methods

**Experimental study design**. The PALOMA-3 study was an international, multicenter, phase III, double-blind randomized controlled trial of palbociclib in women with advanced, estrogen receptor-positive, HER2-negative breast cancer[25, 40]. Post-menopausal patients were required to have received aromatase inhibitor therapy previously, with disease relapse during or within 1 month of treatment for metastatic disease, or during or within 12 months of completing adjuvant treatment. Pre-menopausal or peri-menopausal women were required to have relapsed during or within 1 month of endocrine treatment for metastatic disease, or during or within 12 months of completing adjuvant tamoxifen. Pre-menopausal or peri-menopausal women also received goserelin. The study enrolled 521 women, randomizing in a 2:1 ratio to receive palbociclib plus fulvestrant or fulvestrant plus placebo. Participants received either 125 mg daily of oral palbociclib for 3 weeks

followed by a 7 day treatment break or matching placebo. All patients received 500 mg fulvestrant delivered intramuscularly every 2 weeks for three injections, followed by every 4 weeks. The ctDNA study reported here was conducted on plasma samples collected prospectively for ctDNA analysis. The primary aim of this study was to assess ctDNA mutations and dynamics as a biomarker of progression-free survival, and to compare the dynamics of *PIK3CA* and *ESR1* mutations. Sample size was determined by availability of samples from the clinical trial.

**Plasma collection and processing.** Plasma samples were collected at day 1, day 15, and end of treatment in EDTA blood collection tubes. Samples were processed within 30 min of collection by centrifugation at $1500–2000 \times g$ for 10 min. Plasma was then separated and stored at $-70$ to $-80\,°C$. Prior to DNA extraction plasma samples were thawed and subjected to further centrifugation at $3000 \times g$ for 10 min.

**DNA extraction.** The Circulating Nucleic Acid kit (Cat No./ID: 55114) from Qiagen® (Venlo, The Netherlands) was used according to the manufacturer's instructions. Briefly, plasma was incubated with proteinase K and ACL buffer with added carrier RNA dissolved in AVE buffer for 30 min at 60 °C before ACB buffer was added and the sample passed through a QiaAmp Mini Column. After washes with ACW1, ACW2, and 100% ethanol the column was centrifuged at $20,000 \times g$ for 3 min, dried, and buffer AVE used for elution. The DNA was stored at $-20\,°C$ until analysis. All subsequent analyses used the first elution.

**Droplet digital PCR.** DNA concentration was estimated in each sample by assessment with a TaqMan™ Copy Number Reference Assay (4403326, Thermo Fisher, Massachusetts,USA) against *RPPH1*, the gene-encoding RNAseP. Using the Bio-Rad® AutoDG® automated droplet generator, samples were partitioned into approximately 20,000 micelles in emulsion and PCR performed. The wells were analyzed on a Bio-Rad® QX200 instrument for FAM and VIC signal in each droplet. The Bio-Rad QuantaSoft® software package version 1.4.0.99 was used fit the proportion of empty droplets to a Poisson distribution to calculate the original concentration of FAM and VIC-binding template. Copies/ml and allele fraction were calculated directly from this.

A *PIK3CA* multiplex droplet digital PCR assay was used to assess E542K (c.1624 G > A), E545K (c.1633 G > A), H1047R (c.3140 A > G), and H1047L (c.3140 A > T). Samples were analyzed using either 0.25 ml plasma equivalent or 1.3 ng, whichever was the larger, the latter corresponding to approximately 400 wild-type alleles. A multiplex was called as mutation positive if there were at least two FAM-positive droplets, at least one of these being FAM only. Samples were only called as undetectable if there were at least 300 wild-type alleles detected. If this criterion was not met then the sample was repeated or designated as failed if there was insufficient material to repeat. Individual mutations were subsequently confirmed by repeating the same plasma equivalent with a singleplex assay. As the mutation had previously been identified in the multiplex a single FAM-positive droplet was accepted as positive. All mutation assays were run with three separate non-template controls, with no positive calls made above the control FAM signal.

*ESR1* mutations were interrogated using two separate droplet digital PCR multiplexes from Bio-Rad®, multiplex 1 (dHsaMDXE91450042) and multiplex 2 (dHsaMDXE65719815), updating and expanding a previous analysis[21]. Multiplex 1 assessed E380Q (c.1138 G > C), L536R (c.1607 T > G), Y537C (c.1610 A > G), D538G (c.1613 A > G) and multiplex 2 assessed S463P (c.1387 T > C), Y537N (c.1609 T > A), and Y537S (c.1610 A > C). Mutations were then confirmed in singleplex as for the *PIK3CA* mutations.

Matched day 15 and end of treatment plasma samples were assessed for mutations detected at baseline in *PIK3CA* and *ESR1* with singleplex assays under the same conditions as described above, with the same plasma equivalents tested as previously for end of treatment but 0.5 ml equivalent/1.3 ng, whichever the larger, for day 15. The same criteria was used as for the baseline samples for a sample being called as undetectable. Given the increasing error in estimates from digital PCR at very low concentrations of template, only those samples with four or more FAM-positive droplets in the baseline mutation assay were taken forward for longitudinal analysis.

**The Cancer Genome Atlas.** The Cancer Genome Atlas data for the *PIK3CA* mutations E542K, E545K, H1047R, and H1047L and the *ESR1* mutations E380Q, L536R, Y537C, D538G, S463P, Y537N, and Y537S in breast cancer were accessed electronically from cBioPortal[28, 29] (http://cbioportal.org).

**Statistical analysis.** The "circulating tumor DNA response" was defined as the ratio of mutant copies/ml of plasma at day 15 and day 1, subsequently denoted as $CDR_{15}$. Aggregate totals of *PIK3CA* and *ESR1* mutation copies/ml were used where a sample exhibited polyclonality for these mutations. Paired day 1 and day 15 copies/ml and allele fractions were compared using Wilcoxon signed-rank test. The distribution of copies/ml between *PIK3CA* and *ESR1* were compared using the Mann–Whitney test. Comparisons of $CDR_{15}$ distributions were treated similarly. Harrell's c-index was used to optimize the $CDR_{15}$ cut-off. The test for an association between undetectable mutation at day 15 and end of treatment was conducted with Fisher's exact test. Proportions of patients with undetectable mutations at the end of treatment for *PIK3CA* and *ESR1* were compared with a two sample test of proportions. For all analyses performed a *p* value of <0.05 was considered statistically significant.

For analyses of PFS, Kaplan–Meier curves were plotted and groups compared using the log-rank test. Hazard ratios and associated 95% CIs were obtained from Cox proportional hazards regression models. In order to determine the optimal cut-point for $CDR_{15}$ in relation to PFS, separate models were fitted with data cut at each observed value of $CDR_{15}$ with the cut-point giving the highest Harrell's c-index considered optimal.

**Data availability**. The data that support the findings of this study are available from the corresponding author upon reasonable request.

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

## Acknowledgements

We thank the patients, families, and trial staff who took part in the PALOMA-3 trial. This research was funded by The Medical Research Council (MR/N002121/1), Breast Cancer Now with support from the Mary-Jean Mitchell Green Foundation, Le Cure, and Pfizer. ICR-CTSU receives program grant funding from Cancer Research UK (grant C1491/A15955). We acknowledge National Institute for Health Research funding to the Royal Marsden and Institute of Cancer Research Biomedical Research Centre.

## Author contributions

C.H.B., M.K., M.C., and N.C.T. conceived and designed the PALOMA-3 study. B.O'L., S.H., I.G.-M., C.H.B, M.K., M.C., and N.C.T. conceived and designed the project. B.O'L., S.H., M.B., J.P.M., C.F., Y.L., I.G.-M., C.H.B., M.K., M.C., and N.C.T. collected and assembled the data. B.O'L., S.H, J.P.M., X.H., Y.L., I.G.-M., C.H.B., M.K., M.C., J.M.B., and N.C.T. undertook data analysis and interpretation. B.O'L., S.H., J.P.M., M.B., C.F., Y. L., C.H.B., M.K., M.C., I.G.-M., J.M.B., and N.C.T. wrote the manuscript.

## Additional information

**Competing interests:** B.O'L. received research funding from Pfizer (Inst). M.B. purchased stock from Randox Laboratories. X.H., Y.L., C.H.B., and M.K. are Pfizer employees and have Pfizer stock. M.C. received honoraria from Agendia, Dompe Farmaceutici, Celgene, Pfizer and has a consulting or advisory role in Dompe Farmaceutici, Cynvenio Biosystems, and Newomics. J.M.B. received research funding from AstraZeneca (Inst), Pfizer (Inst), Janssen Cilag (Inst), Novartis (Inst), Roche (Inst), and Clovis Oncology (Inst). N.C.T. has a consulting or advisory role in Roche, Pfizer, Novartis, AstraZeneca, and received research funding from Pfizer (Inst), Roche (Inst), and AstraZeneca. The remaining authors declare no competing financial interests.

