## [Peer Review File · Nature Communications]

Reviewers' comments:

Reviewer #1 (Remarks to the Author):

Early circulating tumor DNA dynamics and clonal selection with palbociclib and fulvestrant for advanced breast cancer

O'leary et al. present a methodologically robust analysis of plasma samples collected within the context of a phase III clinical trial investigating the combination of palbociclib and fulvestrant for advanced breast cancer. Of interest is the finding that circulating tumor DNA dynamics 15 days following commencement of anti-proliferative treatment has potential to predict progression free survival. The authors use the available data from the PALOMA study to determine a threshold for change in circulating tumor DNA concentration 15 days following commencement of palbociclib that best predicts relapse free survival. This work is novel and could be validated in a larger cohort as a marker of palbociclib response. If validated this approach could be used as a pharmacodynamic marker for early-phase studies of cytostatic therapies. The finding that assumed clonal variants are useful prognostic indicators in contrast to subclonal variants which do not provide prognostic information is also helpful for the field.

There are however caveats to the study which largely centre around discussion of clonality and subclonality based on plasma variant allele frequency – this is transparent and highlighted in the discussion and text. The inferences made surrounding ESR status and response to treatment require elaboration and the authors should highlight in the introduction that they were limited in terms of the quantity of plasma/cell free DNA they could analyse compared to other ctDNA studies (0.25ml plasma or 1.3ng cell-free DNA) which could effect limit of detection.

Major

PIK3CA clonality

The authors track variants in the PIK3CA gene on the basis that these variants are predominantly truncal in breast cancer, furthermore they use the likely truncal nature of PIK3CA to suggest that ESR variants are largely subclonal. Based on available data (e.g. TCGA) could the authors describe and quantify the frequency of subclonal PIK3CA variants in the patient demographic analysed in this study and document this in the text? This is important to interpret the claims made in the manuscript.

Comparing CDR15 between clonal and subclonal variants considering assay limit of detection

The authors draw attention to the fact that variants in ESR1 are typically subclonal whereas PIK3CA is clonal and that the summed MAF for PIK3CA variants is more than that for ESR1 variants. Given that the CDR is a ratio of day 1 copies per ml to day 15 copies per ml and given that the ESR1 variants are subclonal and consequently exhibit lower starting MAF - are they more likely to fall below limit of detection of the ddPCR assay than PIK3CA variants? If so is the comparison between subclonal and clonal CDR15 ratios robust given that subclonal variants are more likely to exhibit CDR15s of 0 by falling below LOD? This question is applicable to data presented in figure 2e (potentially figure 2d but it appears no CDR15 fell to 0) and Figure 5b? Could this limitation have effected the lack of prognostic information conferred by ESR1 CDR15 in figure 8? The authors opinion on this would be very helpful.

Since there was a linear relationship between CDR15 with both PIK3CA and ESR1 in most of the cases with dual PIK3CA and ESR1 variants (figure 4d) can the authors analyse the cases where ESR1 CDR15 fell to 0 and determine what the expected mutant ESR1 copies per ml would have been in these 6 cases if a linear relationship had existed here as well. Using this data and considering cell free DNA genomic equivalents analysed at D15 in these cases - what is the

likelihood ESR1 would have been detected within the context of a linear fall in ESR1 CDR15 i.e. is this a true non-linear fall in ESR1 levels at day 15 in these 6 cases or a LOD issue at low MAFs given cfDNA genomic equivalents analysed?

Minor

It is difficult to see the median lines on the dot-plots presented in supplementary Figure 2 and 5

It is interesting that wild type PIK3CA/ESR molecules also decreased in response to treatment at day 15. Was this only in the Palbociclib treated group? Could the authors possibly comment on potential reasons why this would be observed?

Could the authors describe the possible clinical role for using ctDNA within the context they outline within the manuscript. Would they consider cessation of treatment at day 15 for patients without evidence of a ctDNA response to the therapeutic? Or do they envisage use of this technology in early-phase studies as a pharmacodynamic marker of response to novel agents?

Could the authors make it clearer that the survival analysis (using Harrell's C) presumably constitutes a training exercise and therefore requires a validation cohort to ascertain the derived CDR15 cut-off's utility in a predictive context?

Can the authors provide supplementary data regarding cell-free DNA quantity (ng) extracted and analysed for each patient at each time point?

What do the two bar charts in supplementary figure 4 represent, can the legend be clearer?

Reviewer #2 (Remarks to the Author):

O'Leary et al describe an interesting analysis of mutant copies of PIK3CA (often clonal) and ESR1 (often subclonal) in cell-free DNA after 2 weeks of treatment with fulvestrant +/- palbociclib in the PALOMA-3 trial. They find that greater decrease in mutant copies of PIK3CA at this time point is prognostic for improved progression-free survival. They also find greater decrease of mutant copies of ESR1 (relative to PIK3CA) in patients treated with fulvestrant alone, but this decrease did not predict improved progression-free survival; they postulate that ESR1 subclones may be more sensitive to fulvestrant than the overall tumor, but being subclonal, this differential sensitivity may have little impact on patient outcome. Overall, this work adds significant value to the growing literature supporting using ctDNA metrics to prognosticate and predict response in advanced cancer.

Major Comments

The biggest question that emerged from this study that remains unanswered is whether CDR15 would add value to other clinical predictors of PFS, and whether it would be a better predictor than baseline copies of mutant PIK3CA (a measure not reported in this paper). I would like to see a multivariate analysis including baseline copies of PIK3CA, CDR15 of PIK3CA, disease-free interval, number of previous lines of endocrine treatment, and visceral vs bone-only disease. If CDR15 loses significance in such an analysis, this would be important to know and would temper enthusiasm over its future clinical utility. Nonetheless, understanding the underlying biology would be important. Presumably CDR15 correlates with baseline copies (e.g. it is easier to have a lower ratio if you started higher) – could it be baseline copies that matters? Alternatively, does CDR15 (and/or baseline copies) correlate with other important predictors of disease aggressiveness as outlined in clinical predictors above (e.g. one could imagine that ctDNA dynamics might be dependent on site of metastatic disease)? These are clinically relevant questions that could be explored with the existing data.

Minor Comments

Abstract

- It is misleading to say used plasma samples from 455 patients for this analysis, as for the main results/conclusions only used patients who had the tested mutation as well as matched plasma samples at baseline and cycle 1 day 15. Would therefore report in the abstract N=73 (PIK3CA) and N=65 (ESR1).
- Should say explicitly what about PIK3CA ctDNA dynamics specifically predicts what outcome in the abstract – i.e. greater relative decrease in number of mutant PIK3CA copies on day 15 of treatment predicts increased progression-free survival (rather than “PIK3CA ctDNA dynamics after 15 days treatment predicts outcome”). I also find it misleading to use the hazard ratio of 4.92 here, given that that was from the discovery cohort, and would either divide the cohort into discovery and validation and report HR from validation cohort, or simply the median CDR15 as the divider (see comments on results).

Results

- Section 1: How did you determine that the second PIK3CA mutation in the 4 cases with 2 mutations was subclonal vs clonal? Was the VAF for the second mutation significantly lower than for the first in all 4 cases?
- Section 3: Please report what the tertiles (and median) of CDR15 were.
- Section 3: Reasonable to report what the “ideal” cut-off was per supplementary figure 7, but only reasonable to report median PFS in each group if you are using a validation (not discovery) cohort. So either divide the cohort into discovery (to find the cut-off) and validation (to find the median PFS in each group), or just report median PFS for above and below the median. Would change Fig 3B accordingly (i.e. could present both discovery and validation cohorts’ K-M curves, or change to K-M curve based on median). See comments on abstract.
- Section 3: I am puzzled why you did not perform the same tertile and median K-M analysis of the patients treated with fulvestrant alone – there were patients treated with fulvestrant alone who had a CDR15 > 1, and others who had a CDR15 < 0.5, so it’s not empirically obvious why there would be no differentiation there. Should report that there was no difference by tertile or median for patients treated with fulvestrant alone if that is the case.

Figures

- Fig 1A is cut off at the top
- Set x-axis of Fig 1C to 1.0 (i.e. bars below the x-axis would then reflect decrease in ctDNA and bars above would reflect increase, like a waterfall plot)
- Remove second component of Fig 1C since it is presented in Fig 2D/E anyway (to avoid confusion).

We thank the reviewers for highly constructive comments on the manuscript. We have addressed these in full. We provide a point by point rebuttal of the reviewers' comments below.

Reviewer #1 (Remarks to the Author):

Early circulating tumor DNA dynamics and clonal selection with palbociclib and fulvestrant for advanced breast cancer

O'leary et al. present a methodologically robust analysis of plasma samples collected within the context of a phase III clinical trial investigating the combination of palbociclib and fulvestrant for advanced breast cancer. Of interest is the finding that circulating tumor DNA dynamics 15 days following commencement of anti-proliferative treatment has potential to predict progression free survival. The authors use the available data from the PALOMA study to determine a threshold for change in circulating tumor DNA concentration 15 days following commencement of palbociclib that best predicts relapse free survival. This work is novel and could be validated in a larger cohort as a marker of palbociclib response. If validated this approach could be used as a pharmacodynamic marker for early-phase studies of cytostatic therapies. The finding that assumed clonal variants are useful prognostic indicators in contrast to subclonal variants which do not provide prognostic information is also helpful for the field.

There are however caveats to the study which largely centre around discussion of clonality and subclonality based on plasma variant allele frequency – this is transparent and highlighted in the discussion and text. The inferences made surrounding ESR status and response to treatment require elaboration and the authors should highlight in the introduction that they were limited in terms of the quantity of plasma/cell free DNA they could analyse compared to other ctDNA studies (0.25ml plasma or 1.3ng cell-free DNA) which could affect limit of detection.

We thank the reviewer for their positive comments and recommendations for expansion. As suggested we have enriched our analysis of the TCGA data for polyclonal *PIK3CA* mutations and performed further experiments to further establish the potential role the assay limit of detection may have played in our results. These are detailed point by point below.

Major

PIK3CA clonality

The authors track variants in the *PIK3CA* gene on the basis that these variants are predominantly truncal in breast cancer, furthermore they use the likely truncal nature of *PIK3CA* to suggest that ESR variants are largely subclonal. Based on available data (e.g. TCGA) could the authors describe and quantify the frequency of subclonal *PIK3CA*

variants in the patient demographic analysed in this study and document this in the text? This is important to interpret the claims made in the manuscript.

We accessed the TCGA via cBioportal and downloaded the 2015 TCGA, isolating the ER+/HER2- subset as most comparable to the patient cohort in PALOMA-3. Examining the data for the 4 *PIK3CA* mutations assayed in our study we find that multiple *PIK3CA* mutations are rare in primary disease (1/488, 0.2%, accounting for 1/152, 0.7% of all H1047R, H1047L, E542K, E545K mutations), supportive of the hypothesis that *PIK3CA* mutations are generally truncal. These data are now detailed in the results section and we return to it in the discussion but caveat this observation with the fact that the landscape of *PIK3CA* mutations in endocrine-resistant disease is less well-described. The cases of multiple *PIK3CA* mutations are too small in number in both the TCGA and PALOMA-3 to enable meaningful statistical analysis. We now include a detailed breakdown of the observed PALOMA-3 polyclonal *PIK3CA* mutations as supplementary figure 2.

Comparing CDR15 between clonal and subclonal variants considering assay limit of detection

The authors draw attention to the fact that variants in *ESR1* are typically subclonal whereas *PIK3CA* is clonal and that the summed MAF for *PIK3CA* variants is more than that for *ESR1* variants. Given that the CDR is a ratio of day 1 copies per ml to day 15 copies per ml and given that the *ESR1* variants are subclonal and consequently exhibit lower starting MAF - are they more likely to fall below limit of detection of the ddPCR assay than *PIK3CA* variants? If so is the comparison between subclonal and clonal CDR15 ratios robust given that subclonal variants are more likely to exhibit CDR15s of 0 by falling below LOD? This question is applicable to data presented in figure 2e (potentially figure 2d but it appears no CDR15 fell to 0) and Figure 5b? Could this limitation have effected the lack of prognostic information conferred by *ESR1* CDR15 in figure 8? The authors opinion on this would be very helpful.

We have reanalysed the data to investigate whether the analysis presented in Figure 2e is affected by baseline level, i.e. to address whether *ESR1* mutations are more likely have a CDR₁₅s of 0, as they frequently start at a lower baseline level. There is no difference in the baseline *ESR1* mutation level between those patients that had a CDR₁₅ of 0, and those with CDR₁₅ above zero. Therefore there was no suggestion that the increased suppression of *ESR1* mutations observed in figure 2e is due to LOD issues, Conversely, we do observe a statistically significant difference for *PIK3CA* (p=0.029 Mann-Whitney, now supplementary figure 16).

We have also reanalysed the data to demonstrate that baseline mutant copies/ml was not correlated with CDR_{15} , data now included as supplementary figure 10 (*PIK3CA* Spearman's $r = -0.17$, 95%CI $-0.39 - 0.07$, $p=0.16$)(*ESR1* Spearman's $r = 0.13$, 95%CI $-0.13 - 0.37$, $p=0.30$), suggesting CDR_{15} is not a surrogate for baseline mutant copies/ml.

Overall these analyses demonstrate convincingly that the differences observed in figure 2e are not due to analytical issues around detecting mutations, but are due to biological differences. As there is no evidence that analytical issues affect *ESR1* CDR_{15} , this would not limit the potential for *ESR1* CDR_{15} to predict PFS as shown in supplementary figure 9.

Since there was a linear relationship between CDR15 with both PIK3CA and ESR1 in most of the cases with dual PIK3CA and ESR1 variants (figure 4d) can the authors analyse the cases where ESR1 CDR15 fell to 0 and determine what the expected mutant ESR1 copies per ml would have been in these 6 cases if a linear relationship had existed here as well. Using this data and considering cell free DNA genomic equivalents analysed at D15 in these cases - what is the likelihood ESR1 would have been detected within the context of a linear fall in ESR1 CDR15 i.e. is this a true non-linear fall in ESR1 levels at day 15 in these 6 cases or a LOD issue at low MAFs given cfDNA genomic equivalents analysed?

We thank the reviewer for raising this excellent point.. To investigate the possible of effect limit of detection we analysed further day 15 plasma. There are 8 cases, not 6, two pairs lie on top of each other, now corrected in the text. Of these 8 samples, 4/8 cases are above the level of detection and are a true non-linear fall, whilst 4/8 are below the level of detection. For the samples with *ESR1* below the limit of detection we have analysed further plasma, to bring *ESR1* within the lower the limit of detection. 6 mutations were tested, as 2 of the 4 cases were polyclonal. An *ESR1* mutation was detected in only one of these cases, suggesting that falling below the limit of detection is not a major limiting factor in this analysis. These additional analyses are now included in the results, with the below tables included as supplementary figures 13 and 14. The theoretical limit of detection calculation is included in the methods. The issue of limit of detection has also been included in the discussion to emphasise caution in the subclonality observations.

Lab ID	D1 PIK3CA total mutant copies/ml	D15 PIK3CA total mutant copies/ml	D15/D1 PIK3CA total mutant copies/ml	D1 ESR1 total mutant copies/ml	D15 ESR1 total mutant copies/ml	D15/D1 ESR1 total mutant copies/ml	Inferred ESR1 mutant copies/ml	Below LOD
253	3708	7	0.002	886	0	0	2	TRUE
152	2751	171	0.062	415	0	0	26	FALSE
132	1584	117	0.074	27	0	0	2	TRUE
241	555	44	0.078	67	0	0	5	FALSE
72	542	9	0.016	212	0	0	3	TRUE
27	152	97	0.643	1414	0	0	909	FALSE
323	121	5	0.041	52	0	0	2	TRUE
85	58	5	0.088	55	0	0	5	FALSE

Lab ID	Inferred ESR1 mutant copies/ml	New LOD with extra volume analysed	D538G copies/ml	E380Q copies/ml	D15 ESR1 total mutant copies/ml	D15/D1 ESR1 total mutant copies/ml	Change in status
253	2	2	0	0	0	0	FALSE
132	2	2	N/A	0	0	0	FALSE
72	3	2	5	0	5	0.024	TRUE
323	2	2	0	N/A	0	0	FALSE

Minor

It is difficult to see the median lines on the dot-plots presented in supplementary Figure 2 and 5

We have corrected this.

It is interesting that wild type *PIK3CA*/*ESR1* molecules also decreased in response to treatment at day 15. Was this only in the Palbociclib treated group? Could the authors possibly comment on potential reasons why this would be observed?

We have now included an analysis of CDR₁₅ for the wild type *PIK3CA* allele in the patients with *PIK3CA* mutations by treatment. The effect was predominantly seen in the palbociclib group (Supplementary figure 4. Correlation analyses of the absolute reduction of mutant and wild type copies/ml (*ESR1* Spearman's $r = 0.57$, 95%CI 0.38 – 0.72) $p < 0.0001$ *PIK3CA* Spearman's $r = 0.58$ (95%CI 0.40 – 0.72, $p < 0.0001$) suggest at least some of this effect is reduced wild type allele from the tumor, although the cytostatic effect of palbociclib on hemopoietic cells could also be contributing.

Could the authors describe the possible clinical role for using ctDNA within the context they outline within the manuscript. Would they consider cessation of treatment at day 15 for patients without evidence of a ctDNA response to the therapeutic? Or do they envisage use of this technology in early-phase studies as a pharmacodynamic marker of response to novel agents?

The optimal way to use CDR₁₅ in the clinic will be the subject of future trials. We plan trials of the addition of extra treatment in patients without a ctDNA response. We agree that ctDNA response could be used to screen agents in early phase trials, as a potentially highly robust surrogate endpoint. These points have been expanded in the discussion.

Could the authors make it clearer that the survival analysis (using Harrell's C) presumably

constitutes a training exercise and therefore requires a validation cohort to ascertain the derived CDR15 cut-off's utility in a predictive context?

We highlight this in the discussion.

Can the authors provide supplementary data regarding cell-free DNA quantity (ng) extracted and analysed for each patient at each time point?

This is now attached as a supplementary datasheet Excel file.

What do the two bar charts in supplementary figure 4 represent, can the legend be clearer?

We apologise for not making the legend clear, and have clarified in the legend. Each bar represents a single patient. These illustrate the contribution of each of the individual *ESR1* mutations to the total *ESR1* mutant copies/ml in those patients who in the baseline samples were identified as having more than one *ESR1* mutation. The lower panel is an enlargement of the upper to enable a clearer look at the patients with lower mutation abundance.

Reviewer #2 (Remarks to the Author):

O'Leary et al describe an interesting analysis of mutant copies of PIK3CA (often clonal) and ESR1 (often subclonal) in cell-free DNA after 2 weeks of treatment with fulvestrant +/- palbociclib in the PALOMA-3 trial. They find that greater decrease in mutant copies of PIK3CA at this time point is prognostic for improved progression-free survival. They also find greater decrease of mutant copies of ESR1 (relative to PIK3CA) in patients treated with fulvestrant alone, but this decrease did not predict improved progression-free survival; they postulate that ESR1 subclones may be more sensitive to fulvestrant than the overall tumor, but being subclonal, this differential sensitivity may have little impact on patient outcome. Overall, this work adds significant value to the growing literature supporting using ctDNA metrics to prognosticate and predict response in advanced cancer.

We thank the reviewer for their positive assessment and address their concerns point by point below.

Major Comments

The biggest question that emerged from this study that remains unanswered is whether CDR15 would add value to other clinical predictors of PFS, and whether it would be a better predictor than baseline copies of mutant PIK3CA (a measure not reported in this paper). I would like to see a multivariate analysis including baseline copies of PIK3CA, CDR15 of PIK3CA, disease-free interval, number of previous lines of endocrine treatment, and visceral vs bone-only disease. If CDR15 loses significance in such an analysis, this would be important to know and would temper enthusiasm over its future clinical utility. Nonetheless, understanding the underlying biology would be important. Presumably CDR15 correlates with baseline copies (e.g. it is easier to have a lower ratio if you started higher) – could it be baseline copies that matters? Alternatively, does CDR15 (and/or baseline copies) correlate with other important predictors of disease aggressiveness as outlined in clinical predictors above (e.g. one could imagine that ctDNA dynamics might be dependent on site of metastatic disease)? These are clinically relevant questions that could be explored with the existing data.

We thank the reviewer for these points that we have addressed in full

Baseline mutant copies/ml did not correlate with the CDR₁₅, (*PIK3CA* Spearman's $r = -0.17$, 95%CI $-0.39 - 0.07$, $p=0.16$)(*ESR1* Spearman's $r = 0.13$, 95%CI $-0.13 - 0.37$, $p=0.30$). Furthermore, baseline mutant copies/ml was not predictive of PFS (above vs below median HR=1.22, 95% CI 0.606-2.43, $p=0.582$), These data demonstrate the importance of dynamic assessment of ctDNA in predicting sensitivity to palbociclib. These data are now presented in the results section and have are presented in supplementary figure 10.

D1 versus CDR₁₅

PIK3CA

Spearman's R = -0.17 (95%CI -0.39 – 0.07) p=0.16

ESR1

Spearman's R = 0.13 (95%CI -0.13 – 0.37) p=0.30

We have performed the suggested multivariate analysis which is included as supplementary figure 11 and demonstrates that CDR₁₅ remains a significant predictor when other statistically significant clinical factors are accounted for in the model.

Univariate analysis	Hazard ratio	Lower 95% CI	Upper 95% CI	p value
Prior hormonal therapy (Yes/No)	0.778	0.288	2.1	0.62
Baseline copies/ml mutant PIK3CA	1.22	0.606	2.43	0.582
Site of metastatic disease (Visceral/Non-visceral)	1.69	0.763	3.73	0.191
Menopausal status (Pre/peri v Post)	0.572	0.225	1.46	0.236
Number of prior therapies (1 v >1)	3.83	0.9	16.3	0.0502
Number of disease sites (1 v >1)	2.68	1.11	6.48	0.0234
Disease site liver (Yes v No)	3.39	1.5	7.64	0.00181
CDR15 (High v Low)	4.92	1.98	12.3	0.000178
Multivariate analysis				
Disease site liver (Yes v No)	4.01	1.76	9.15	0.00095
CDR15 (High v Low)	5.73	2.26	14.51	0.00023

Univariate and multivariate analyses

Minor Comments

Abstract

- It is misleading to say used plasma samples from 455 patients for this analysis, as for the main results/conclusions only used patients who had the tested mutation as well as

matched plasma samples at baseline and cycle 1 day 15. Would therefore report in the abstract N=73 (PIK3CA) and N=65 (ESR1).

We have removed the "455" from the abstract as requested.

-Should say explicitly what about PIK3CA ctDNA dynamics specifically predicts what outcome in the abstract – i.e. greater relative decrease in number of mutant PIK3CA copies on day 15 of treatment predicts increased progression-free survival (rather than "PIK3CA ctDNA dynamics after 15 days treatment predicts outcome").

This has now been changed in accordance with the advice from the reviewer and now reads "we show that relative change in *PIK3CA* ctDNA level after 15 days treatment strongly predicts PFS on palbociclib and fulvestrant (hazard ratio 3.94, p = 0.0013). "

I also find it misleading to use the hazard ratio of 4.92 here, given that that was from the discovery cohort, and would either divide the cohort into discovery and validation and report HR from validation cohort, or simply the median CDR15 as the divider (see comments on results).

As requested, we now report the data according to the median in the abstract, and lead with this in the results section.

Results

-Section 1: How did you determine that the second PIK3CA mutation in the 4 cases with 2 mutations was subclonal vs clonal? Was the VAF for the second mutation significantly lower than for the first in all 4 cases?

The second VAF was significantly lower in all cases. These data are now presented in full as part of supplementary figure 2.

-Section 3: Please report what the tertiles (and median) of CDR15 were.

The median CDR₁₅ is now reported in the text for *PIK3CA* and *ESR1* for the palbociclib and fulvestrant patients, and in the relevant supplementary figure legends for the placebo and fulvestrant patients.

-Section 3: Reasonable to report what the “ideal” cut-off was per supplementary figure 7, but only reasonable to report median PFS in each group if you are using a validation (not discovery) cohort. So either divide the cohort into discovery (to find the cut-off) and validation (to find the median PFS in each group), or just report median PFS for above and below the median. Would change Fig 3B accordingly (i.e. could present both discovery and validation cohorts’ K-M curves, or change to K-M curve based on median). See comments on abstract.

As mentioned in an earlier response, the data are now presented according to the median with further clarification highlighting this as a training set.

-Section 3: I am puzzled why you did not perform the same tertile and median K-M analysis of the patients treated with fulvestrant alone – there were patients treated with fulvestrant alone who had a CDR15 > 1, and others who had a CDR15 < 0.5, so it’s not empirically obvious why there would be no differentiation there. Should report that there

was no difference by tertile or median for patients treated with fulvestrant alone if that is the case.

We have performed the requested analysis for fulvestrant plus placebo patients, with KM curves by median displayed in supplementary figures 12 and 13. There are relatively few patients treated with fulvestrant alone.

Supplementary figure 12 (left) Kaplan Meier plot for PFS of patients randomized to placebo and fulvestrant split by median *PIK3CA* CDR₁₅. (Left). Hazard ratio for >median compared with <median = 2.54 (95% CI 0.89 – 7.25). Logrank test p=0.07. Supplementary figure 13 (right). Kaplan Meier plot for PFS of patients randomized to placebo and fulvestrant split by median *ESR1* CDR₁₅. Hazard ratio for >median compared with <median = 2.28 (95% CI 0.77 – 6.70). Logrank test p=0.12.

Figures

-Fig 1A is cut off at the top

We thank the reviewer for pointing this out and have corrected it.

-Set x-axis of Fig 1C to 1.0 (i.e. bars below the x-axis would then reflect decrease in ctDNA and bars above would reflect increase, like a waterfall plot)

We thank the reviewer for this suggestion but for the *ESR1* plot this makes the data somewhat difficult to interpret visually. To emphasize the fall from the value

of one, we have included a dotted line at $y=1$ on the relevant plots, examples below.

ESR1 CDR₁₅ by treatment

ESR1 CDR₁₅ by treatment

-Remove second component of Fig 1C since it is presented in Fig 2D/E anyway (to avoid confusion).

We have removed the lower panel of figure 1C

Reviewers' comments:

Reviewer #1 (Remarks to the Author):

Early circulating tumor DNA dynamics and clonal selection with palbociclib and fulvestrant for advanced breast cancer

Response to rebuttal comments.

The authors have addressed some of my concerns regarding the revised manuscript. I have a few ongoing comments regarding clonality, limit of detection and the new multivariable analysis: The authors suggest that the PIK3CA variants that they track in plasma are predominantly clonal. They now justify this on the basis that multiple PIK3CA variants are rare in primary disease through use of CBIO portal. Although mildly supportive, it is this reviewer's opinion that this is not the optimal way to determine clonality and should not be presented as such. Are there any additional methods that could be used to computationally analyse TCGA data to establish PIK3CA or ESR1 clonality status in ER+ BRCA or references that could be used to illustrate the authors' claims?

Regarding the limit of detection (LOD) issue the authors present some interesting analyses in supplementary to address this important point and I agree the conclusions regarding figure 2d and the prognostic association of CDR15 and outcomes in the manuscript are likely to be not influenced by technical concerns.

I am mildly concerned regarding the conclusions on the analyses in supplementary figure 13 and 14 since only 2 of 8 cases have an inferred ESR mutant copy number >5 copies. The authors should acknowledge their calculated inferred ESR1 mutant copy per ml will be subject to a degree of error (perhaps by presenting confidence intervals) and that plasma sampling error may occur when dealing with low mutant copy numbers in a sample. Therefore, an alternative interpretation of their data is that 6 of 8 cases have CDRs of 0 due to LOD issues rather than non-linear fall - I would therefore remove the discussion surrounding sentence line 206 "indicating contrasting clonal dynamics of a sub-clonal ESR1 mutation" and suggest that this observation could also be technical.

In the supplementary excel documenting cfDNA quantities could the abbreviations be defined i.e. what is EOT?

For the multivariate analysis requested by the other reviewer how did the authors split CDR15 into high and low, can this be made clear in the methods and legend? Why not input CDR15 as a continuous variable into the regression? The authors write that they retained predictors with $P < 0.1$ in the multivariable analysis yet they seem to exclude "number of disease sites" which had a P value of 0.02? Were assumptions for Cox regression met?

Reviewer #2 (Remarks to the Author):

The authors have adequately addressed the prior comments.

Manuscript NCOMMS-17-18123A

Second round reviewers' comments

We thank the reviewer for their further suggestions regarding the manuscript. These have all been incorporated in full with a point by point guide detailed below.

Reviewers' comments:

Reviewer #1 (Remarks to the Author):

Early circulating tumor DNA dynamics and clonal selection with palbociclib and fulvestrant for advanced breast cancer
Response to rebuttal comments.

The authors have addressed some of my concerns regarding the revised manuscript. I have a few ongoing comments regarding clonality, limit of detection and the new multivariable analysis:

The authors suggest that the *PIK3CA* variants that they track in plasma are predominantly clonal. They now justify this on the basis that multiple *PIK3CA* variants are rare in primary disease through use of CBIO portal. Although mildly supportive, it is this reviewer's opinion that this is not the optimal way to determine clonality and should not be presented as such. Are there any additional methods that could be used to computationally analyse TCGA data to establish *PIK3CA* or *ESR1* clonality status in ER+ BRCA or references that could be used to illustrate the authors' claims?

***PIK3CA* has been shown to be predominantly clonal in multiple prior studies and it is widely accepted to be so in the field. In their analysis of 2,433 breast cancers, Pereira et al (Nature Communications 2016) estimated the cancer cell fraction of *PIK3CA* mutations to be close to 1 in all of their Integrative Clusters, indicative of truncal status within a margin of error. Of the available multiple biopsy studies of sufficient size to corroborate this experimentally, Yates et al examined 50 early breast cancers with multiple region sequencing, identifying 10 cancers with *PIK3CA* mutations only one of which appeared subclonal (Nature Medicine 2015). Furthermore, multiple prior studies have shown close agreement in *PIK3CA* mutations status between paired primary and metastatic breast tumours (e.g. Yates et al Cancer Cell 2017), and between primary tumour and metastatic ctDNA analysis (e.g. Higgins et al Clinical Cancer Research 2012, Baselga et al Lancet Oncology 2017). This high rate of agreement is consistent with *PIK3CA* mutations being predominantly clonal. There are cases of divergence (*PIK3CA* wild type primary with mutant**

recurrence) but they are a minority and entirely consistent with the data presented in this manuscript.

The above papers have now been referenced in the manuscript after the statement:

“a similar proportion to that seen in The Cancer Genome Atlas for these mutations in primary disease (22.1%) 28, 29, *PIK3CA* mutations being predominantly truncal events in breast cancer(above references added).”

Regarding the limit of detection (LOD) issue the authors present some interesting analyses in supplementary to address this important point and I agree the conclusions regarding figure 2d and the prognostic association of CDR15 and outcomes in the manuscript are likely to be not influenced by technical concerns.

I am mildly concerned regarding the conclusions on the analyses in supplementary figure 13 and 14 since only 2 of 8 cases have an inferred *ESR1* mutant copy number >5 copies. The authors should acknowledge their calculated inferred *ESR1* mutant copy per ml will be subject to a degree of error (perhaps by presenting confidence intervals) and that plasma sampling error may occur when dealing with low mutant copy numbers in a sample. Therefore, an alternative interpretation of their data is that 6 of 8 cases have CDRs of 0 due to LOD issues rather than non-linear fall - I would therefore remove the discussion surrounding sentence line 206 “indicating contrasting clonal dynamics of a sub-clonal *ESR1* mutation” and suggest that this observation could also be technical.

We thank the reviewer for these useful clarifications that we have now made in the manuscript. The sentence has been changed to:

“**suggestive** of contrasting clonal dynamics of a sub-clonal *ESR1* mutation (Fig. 4D). Although the *ESR1* mutant clone becoming undetectable **could in part be due to difference in level of detection between *PIK3CA* and *ESR1* mutations** (Supplementary figures 14 and 15), *ESR1* became undetectable at a higher rate than would be expected by random sampling taking into account difference in level of detection (Supplementary table 1).”

To further support our results we performed a statistical simulation to assess the likelihood that selective loss of *ESR1* without loss of *PIK3CA* was due to sampling error – i.e. that for discordantly negative *ESR1* results there was actually no amplifiable template in the assessed volume for each case despite a true linear relationship between the fall in *PIK3CA* and *ESR1* mutation copies. Accordingly we assumed the true level of *ESR1* mutant DNA was implied by the fall in *PIK3CA* for all 25 samples shown in Figure 4D, where 40% (10/25) samples had undetectable *ESR1* mutation at day 15. We then simulated sequential sampling of 0.5ml plasma equivalent for each of the 25 samples

using Poisson distributions implied by the inferred *ESR1* mutant concentrations to observe the number of negative results that might be expected from a random sampling process. The simulation was run 10,000 times and resulted in 10 or more samples with undetectable *ESR1* mutations on 0.1% occasions (10/10,000, Supplementary table 1) implying that plasma sampling was an unlikely overall explanation for the observations.

Number of negative samples out of 25 in single simulation	2	3	4	5	6	7	8	9	10	11
Number of times outcome observed after 10,000 repeats	0	474	1887	2944	2642	1450	484	109	10	0

Supplementary table 1. Simulated data to assess the likelihood of sampling error explaining the number of discordant *ESR1* undetectable tests in day 15 samples assessed for both *ESR1* and *PIK3CA* CDR₁₅ (Figure 4D). To model sampling error we defined the probability of having a positive result for each day15 sample by using the inferred copies/ml of *ESR1*, assuming equivalent CDR₁₅ to *PIK3CA* and independence of sampling events. This probability of *ESR1* being undetectable by sampling error was modelled using an individual Poisson distribution $Po(\lambda)$ for each sample, where λ = inferred concentration and the probability of a negative result arising as a result of sampling error $Po(X=0) = e^{-\lambda}$. The observed rate of undetectable *ESR1* was 10/25 samples (40%, Figure 4D). In the simulation, repeated 10,000 times, the frequency of 10 or more negatives was 0.1% (10/10,000).

In the supplementary excel documenting cfDNA quantities could the abbreviations be defined i.e. what is EOT?

EOT is end of treatment, this has been changed in the Excel sheet where space permits. Other labels have been clarified.

For the multivariate analysis requested by the other reviewer how did the authors split CDR15 into high and low, can this be made clear in the methods and legend? Why not input CDR15 as a continuous variable into the regression? The authors write that they retained predictors with $P < 0.1$ in the multivariable analysis yet they seem to exclude “number of disease sites” which had a P value of 0.02? Were assumptions for Cox regression met?

For the multivariate model CDR₁₅ was split using Harrell's c index as per figure 3C, this has now been made clear in the legend and methods, and is discussed in the results. A high/low split was used in the model as this was the approach taken in the rest of the manuscript. For the multivariate analysis we retained predictors with $p < 0.1$ from the univariate analyses (4 out of 7). Two of the predictors stayed in the final model (met the typical $p < 0.05$ criteria) and other two predictors dropped (with p-value greater than 0.05 when all 4 predictors were included in the multivariable analysis). The assumption for Cox regression met and there was not interaction between the two variables. This detail has been added to the legend.

Reviewer #2 (Remarks to the Author):

The authors have adequately addressed the prior comments.

REVIEWERS' COMMENTS:

Reviewer #1 (Remarks to the Author):

The authors have addressed my prior concerns. I have no further comments.